# Mesospheric nitric oxide model from SCIAMACHY data

Stefan Bender<sup>1</sup>, Miriam Sinnhuber<sup>2</sup>, Patrick J. Espy<sup>1</sup>, and John P. Burrows<sup>3</sup> <sup>1</sup>Norwegian University of Science and Technology, Trondheim, Norway <sup>2</sup>Karlsruhe Institute of Technology, Karlsruhe, Germany <sup>3</sup>University of Bremen, Bremen, Germany

Correspondence: Stefan Bender (stefan.bender@ntnu.no)

Abstract. We present an empirical model for nitric oxide (NO) in the mesosphere ( $\approx 60-90$  km) derived from SCIAMACHY limb scan data. This work complements and extends the NOEM (Nitric Oxide Empirical Model, Marsh et al. (2004)) and SANOMA (SMR Acquired Nitric Oxide Model Atmosphere, Kiviranta et al. (2018)) empirical models in the lower thermosphere. The regression ansatz builds on the heritage of studies by Hendrickx et al. (2017) and the super-posed epoch analysis

by Sinnhuber et al. (2016) which estimate NO production from particle precipitation.

Our model relates the daily (longitudinally) averaged NO number densities from SCIAMACHY (Bender et al., 2017b, a) as a function of geomagnetic latitude to the solar Lyman- $\alpha$  and the geomagnetic AE indices. We use a non-linear regression model incorporating a finite and seasonally varying lifetime for the geomagnetically induced NO. We estimate the parameters by finding the maximum posterior probability and calculate the parameter uncertainties using Markov-Chain Monte-Carlo

sampling. In addition to providing an estimate of the NO content in the mesosphere, the regression coefficients indicate regions where certain processes dominate.

#### 1 Introduction

It has been recognized in the past decades that the mesosphere and stratosphere are coupled in various ways (Baldwin and Dunkerton, 2001). Consequently, climate models have been evolving to extend to increasingly higher levels in the atmosphere

- to improve the accuracy of medium- and long-term predictions. Nowadays it is not unusual that these models include the mesosphere (40 km–90 km) or the lower thermosphere (90 km–120 km) (Matthes et al., 2017). It is therefore important to understand the processes in the mesosphere and lower thermosphere and to find the important drivers of chemistry and dynamics in that region. The atmosphere above about 100 km is coupled to solar and geomagnetic activity, also known as space weather (Sinnhuber et al., 2012). Electrons and protons from the solar wind and the radiation belts with sufficient kinetic energy
- enter the atmosphere in that region. Since as charged particles they move along the magnetic field, this precipitation occurs primarily at high geomagnetic latitudes.

Previously the role of NO in the mesosphere has been identified as an important free radical, and in this sense a driver of the chemistry (Kockarts, 1980; Barth, 1992, 1995; Roble, 1995; Bailey et al., 2002; Barth et al., 2009; Barth, 2010), particularly during winter when it is long lived because of reduced photodissociation. NO generated in the region between 90 km

and 120 km at auroral latitudes is strongly influenced by both solar and geomagnetic activity (Marsh et al., 2004; Sinnhuber

et al., 2011, 2016; Hendrickx et al., 2015, 2017). At high latitudes, NO is transported down to the upper stratosphere during winter, usually down to 50 km and occasionally down to 30 km (Siskind et al., 2000; Randall et al., 2007; Funke et al., 2005a, 2014b). At those altitudes and also in the mesosphere, NO participates in the "odd oxygen catalytic cycle which depletes ozone" (Crutzen, 1970). Additional dynamical processes such as Sudden Stratospheric Warmings (SSW) also result in a strong downward transport of mesospheric air into the upper stratosphere (Pérot et al., 2014; Orsolini et al., 2017).

Different instruments have been measuring NO in the mesosphere and lower thermosphere, but at different altitudes and at different local times. Measurements from solar occultation instruments such as Scisat-1/ACE-FTS or AIM/SOFIE are limited in latitude and local time (sunrise/sunset). Global observations from sun-synchronously orbiting satellites are available from Envisat/MIPAS below 70 km daily and 50 km–200 km every ten days (Funke et al., 2001, 2005b); from Odin/SMR between

- 85 km–115 km (Kiviranta et al., 2018); or from Envisat/SCIAMACHY between 60 km–90 km daily (Bender et al., 2017b) and 60 km–160 km every 15 days (Bender et al., 2013). Because the Odin and Envisat orbits are sun-synchronous, the measurement local times are fixed to around 06:00/18:00 and 10:00/22:00. While MIPAS has both day and night measurements, SCIAMACHY provides day-time (10:00) data because of the measurement principle (fluorescent UV scattering, see Bender et al. (2013, 2017b)). Unfortunately, Envisat stopped communicating in 04/2012 and therefore the data available from MIPAS
- and SCIAMACHY are limited to nearly ten years from 08/2002 to 04/2012. The other aforementioned instruments are still operational and provide ongoing data as long as satellite operations continue.

Chemistry-climate models struggle to simulate the NO amounts and distributions in the mesosphere and lower thermosphere (see, for example, Funke et al. (2017); Randall et al. (2015); Orsolini et al. (2017); Hendrickx et al. (2018)). To remedy the situation, most models parametrize NO by constraining its amount and distribution to measurements. For example, the next

generation of climate simulations (CMIP6, see Matthes et al. (2017)) and other recent model simulations (Sinnhuber et al., 2018) parametrize particle effects as derived partly from Envisat/MIPAS NO measurements.

# NO in the Mesosphere and Lower Thermosphere

NO in the mesosphere and lower thermosphere is produced by  $N_2$  dissociation

$$N_2 + h\nu \to N(^2D) + N(^4S) , \qquad (R1)$$

followed by the reaction of the excited nitrogen atom  $N(^{2}D)$  with molecular oxygen (Solomon et al., 1982; Barth, 1992, 1995):

$$N(^2D) + O_2 \to NO + O . \tag{R2}$$

The binding energy of N<sub>2</sub> is 9.8 eV per bond which sums to about 30 eV for N<sub>2</sub>'s triple bond. This energy (denoted by hν in (R1)) can be provided by a number of sources, most notably by auroral or photoelectrons as well as by soft solar X-rays
(λ < 40 nm).</li>

The NO content is reduced by photodissociation

$$NO + h\nu \rightarrow N + O \quad (\lambda 

by photoionization

 $\mathrm{NO} + h\nu \rightarrow \mathrm{NO}^+ + \mathrm{e}^- \quad (\lambda < 134\,\mathrm{nm}) \; ,$ 

and by reacting with atomic nitrogen:

 $NO + N \rightarrow N_2 + O$ .

- Another effective loss of NO is the photoexcitation of  $NO_x$  (=  $NO + NO_2$ ) and subsequent reactions of excited species which form  $N_2O$  (Maric and Burrows, 1992). The latter acts as an intermediate reservoir at high altitudes ( $\geq 90$  km, see Sheese et al. (2016)), reacting with  $O(^1D)$  in two well known channels to  $N_2$  and  $O_2$  as well as to 2NO. However, the largest  $N_2O$ abundances are located below 60 km and originate primarily from the transport of tropospheric  $N_2O$  into the stratosphere through the Brewer Dobson Circulation (Funke et al., 2008a, b; Sheese et al., 2016) but can reach up to 70 km in geomagnetic
- storm conditions (Funke et al., 2008a; Sheese et al., 2016). Both source and sink reactions indicate that NO behaves differently in sunlit conditions than in dark conditions. NO is produced in dark conditions by particle precipitation at auroral latitudes, but is then depleted only by reacting with atomic nitrogen (reaction (R5)). This asymmetry between production and depletion in dark conditions results in different lifetimes of NO.

Early work to parametrize NO in the lower thermosphere (100 km-150 km) used SNOE measurements from 1999-2001 (Marsh

- et al., 2004). With these three years of data and using empirical orthogonal functions, the so-called NOEM (Nitric Oxide Empirical Model) estimates NO in the lower thermosphere as a function of the solar  $f_{10.7 \text{ cm}}$  radio flux, the solar declination angle, and the planetary Kp index. NOEM is still used as prior input for NO retrieval, for example from MIPAS (Bermejo-Pantaleón et al., 2011; Funke et al., 2012) and SCIAMACHY (Bender et al., 2017b) spectra. However, three years is relatively short compared to the 11-year solar cycle, and the years 1999 to 2001 encompass a period of elevated solar activity. To address
- this, a longer time series from AIM/SOFIE was used to determine the important drivers of NO in the lower thermosphere (90 km–140 km) by Hendrickx et al. (2017). Other recent work uses ten years of NO data from Odin/SMR from 85 km to 115 km (Kiviranta et al., 2018). Funke et al. (2016) derived a semi-empirical model of NO<sub>y</sub> in the stratosphere from MIPAS data. Here we use Envisat/SCIAMACHY NO data from the nominal limb mode (Bender et al., 2017b, a). Apart from providing a similarly long time series of NO data, the nominal Envisat/SCIAMACHY NO data cover the mesosphere from 60 km to 90 km (Bender et al., 2017b), bridging the gap between the stratosphere and lower thermosphere models.
  - The manuscript is organized as follows: we present the data used in this work in Sect. 2. The two model variants, linear and non-linear, are described in Sect. 3. Details about the parameter and uncertainty estimation are explained in Sect. 4, and we present the results in Sect. 5. Finally we conclude our findings in Sect. 6.

(R5)

(R4)

# 2 Data

# 2.1 SCIAMACHY NO

We use the SCIAMACHY (SCanning Imaging Absorption spectroMeter for Atmospheric CHartoghraphY) nitric oxide data set version 6.2.1 (Bender et al., 2017a) retrieved from the nominal limb scan mode ( $\approx 0-93$  km). For a detailed instrument

description, see Burrows et al. (1995); Bovensmann et al. (1999), and for details of the retrieval algorithm, see Bender et al. (2013, 2017b).

The data were retrieved for the whole Envisat period (08/2002–04/2012). This satellite was orbiting in a sun-synchronous orbit at around 800 km altitude, with Equator crossing times of 10:00/22:00 local time. The NO number densities from the SCIAMACHY nominal mode were retrieved from the NO gamma band emissions. Since those emissions are fluorescent

emissions excited by solar UV, SCIAMACHY NO data are only available for the 10:00 dayside (downleg) part of the orbit. Furthermore, the retrieval was carried out for altitudes from 60 km to 160 km, but above approximately 90 km, the data reflect the scaled a priori densities from NOEM (Bender et al., 2017b). We therefore restrict the modelling to the mesosphere below 90 km.

We averaged the individual orbital data longitudinally on a daily basis according to their geomagnetic latitude within  $10^{\circ}$ 

bins. The geomagnetic latitude was determined according to the eccentric dipole approximation of the 12th generation of the International Geomagnetic Reference Field (IGRF12) (Thébault et al., 2015). In the vertical direction the original retrieval grid altitudes (2 km bins) were used.

The measurement sensitivity is taken into account via the averaging kernel diagonal elements, and days where its binned average was below 0.002 were excluded from the timeseries. Considering this criterion, each bin (geomagnetic latitude and altitude) contains about 3400 data points.

# 2.2 Proxies

We use two proxies to model the NO number densities, one accounting for the long-term eleven-year solar cycle and one accounting for the short term geomagnetic activity. Various proxies have been used or proposed to account for the eleven-year solar cycle in mesospheric-thermospheric NO. The NOEM (Nitric Oxide Empirical Model, Marsh et al. (2004)) uses the natural logarithm of the solar 10.7 cm radio flux  $f_{10.7}$ . More recent work on AIM/SOFIE NO (Hendrickx et al., 2017)

- uses the solar Lyman- $\alpha$  index because some of the main production and loss processes are driven by UV photons. Besides accounting for the long-term variation of NO with solar activity, the Lyman- $\alpha$  index also includes short-term UV variations and the associated NO production, for example caused by solar flares. Barth et al. (1988) have shown that the Lyman- $\alpha$  index directly relates to the observed NO at low latitudes (30°S–30°N). Thus we use it in this work as a proxy for NO.
- In the same manner as for the long-term variation, the "right" geomagnetic index to model short-term, particle-induced variations of NO has been a matter of dispute. Kp is the oldest and most commonly used geomagnetic index, it was, for example, used in earlier work by Marsh et al. (2004) for modelling NO in the mesosphere and lower thermosphere. Kp is derived from magnetometer stations distributed at different latitudes and mostly in the northern hemisphere. Its use as a proxy

for NO production has been questioned by Hendrickx et al. (2017), suggesting the auroral electrojet index (AE) (Davis and Sugiura, 1966) as a better choice (see also Sinnhuber et al., 2016). The AE index is derived from stations distributed almost evenly within the auroral latitude band. This distribution enables the AE index to be more closely related to the energy input into the atmosphere at these latitudes. Therefore, we use the auroral electrojet index (AE) as a proxy for geomagnetically induced NO. To account for the 10:00 satellite sampling, we average the hourly AE index from noon the day before to noon on the measurement day.

5

It should be noted that tests using Kp instead of AE and using  $f_{10.7}$  instead of Lyman- $\alpha$  suggested that the particular choice of index did not lead to significantly different results. Our choice of AE rather than Kp and Lyman- $\alpha$  over  $f_{10.7}$  is physically based and motivated as described above.

# 10 3 Regression model

We denote the number density by  $x_{NO}$  as a function of the (geomagnetic) latitude  $\phi$ , the altitude z, and the time (measurement day) t:  $x_{NO}(\phi, z, t)$ . In the following we often drop the subscript NO and combine the time direction into a vector x with the *i*th entry denoting the density at time  $t_i$ , such that  $x_i(\phi, z) = x(\phi, z, t_i)$ .

# 3.1 Linear model

15 In the (multi-)linear case, we relate the nitric oxide number densities  $x_{NO}(\phi, z, t)$  to the two proxies, the solar Lyman- $\alpha$  index (Ly $\alpha(t)$ ) and the geomagnetic AE index (AE(t)). Harmonic terms with  $\omega = 1 a^{-1} = (365.25 d)^{-1}$  account for annual and semiannual variations. The linear model, including a constant offset for the background density, describes the NO density according to Eq. (1):

$$x_{\text{NO}}(\phi, z, t) = a(\phi, z) + b(\phi, z) \cdot \text{Ly}\alpha(t) + c(\phi, z) \cdot \text{AE}(t) + \sum_{n=1}^{2} \left[ d_n(\phi, z) \cos(n\omega t) + e_n(\phi, z) \sin(n\omega t) \right].$$
(1)

20 The linear model can be written in matrix form for the *n* measurement times  $t_1, \ldots, t_n$  as Eq. (2), with the parameter vector  $\beta$  given by  $\beta_{\text{lin}} = (a, b, c, d_1, e_1, d_2, e_2)^{\top} \in \mathbb{R}^7$  and the model matrix  $\mathbf{X} \in \mathbb{R}^{n \times 7}$ . We determine the coefficients via least squares, minimizing the squared differences of the modelled number densities to the measured ones.

## 3.2 Non-linear model

In contrast to the linear model above, we modify the AE index by a finite lifetime  $\tau$  which varies according to season, we denote this modified version by  $\widetilde{AE}$ . We then omit the harmonic parts in the model, and the non-linear model is given by Eq. (3):

$$x_{\rm NO}(\phi, z, t) = a(\phi, z) + b(\phi, z) \cdot \operatorname{Ly}\alpha(t) + c(\phi, z) \cdot \widetilde{\operatorname{AE}}(t) .$$
(3)

$$\boldsymbol{x}_{\text{NO}}(\phi, z) = \begin{pmatrix} 1 & \text{Ly}\alpha(t_1) & \text{AE}(t_1) & \cos(\omega t_1) & \sin(\omega t_1) & \cos(2\omega t_1) & \sin(2\omega t_1) \\ \vdots & & & \vdots \\ 1 & \text{Ly}\alpha(t_n) & \text{AE}(t_n) & \cos(\omega t_n) & \sin(\omega t_n) & \cos(2\omega t_n) & \sin(2\omega t_n) \end{pmatrix} \cdot \begin{pmatrix} a \\ b \\ c \\ d_1 \\ e_1 \\ d_2 \\ e_2 \end{pmatrix}$$
(2)
$$= \mathbf{X} \cdot \boldsymbol{\beta}$$

The lifetime-corrected AE is given by the sum of the previous 60 days' AE values, each multiplied by an exponential decay factor:

$$\widetilde{AE}(t) = \sum_{t_i=0}^{60d} AE(t-t_i) \cdot \exp\left\{-\frac{t_i}{\tau}\right\} .$$
(4)

The total lifetime  $\tau$  is given by a constant part  $\tau_0$  plus the non-negative fraction of a seasonally varying part  $\tau_t$ :

5 
$$au = au_0 + \begin{cases} au_t , & au_t \ge 0 \\ 0 , & au_t 

Table 1. Parameter search space for the non-linear model and uncertainty estimation.

| parameter                     | lower bound               | upper bound            | prior form |
|-------------------------------|---------------------------|------------------------|------------|
| offset (a)                    | $-10^{10}  {\rm cm}^{-3}$ | $10^{10}{\rm cm}^{-3}$ | flat       |
| Lyman- $\alpha$ amplitude (b) | $-10^{10}{\rm cm}^{-3}$   | $10^{10}{\rm cm}^{-3}$ | flat       |
| AE amplitude (c)              | $0\mathrm{cm}^{-3}$       | $10^{10}{\rm cm}^{-3}$ | flat       |
| $	au_0$                       | 0 d                       | 100 d                  | exp        |
| $\tau$ cosine amplitude (d)   | $-100\mathrm{d}$          | 100 d                  | exp        |
| au sine amplitude (e)         | $-100\mathrm{d}$          | 100 d                  | exp        |

# 4.1 Maximum posterior probability

Because of the complicated structure of the model function Eq. (3), in particular the lifetime parts in Eqs. (5) and (6), the usual gradient methods converge slowly, if at all. Therefore, we fit the parameters and assess their uncertainty ranges using Markov-Chain Monte-Carlo (MCMC) sampling (Foreman-Mackey et al., 2013). This method samples probability distributions and we apply it to sample the parameter space putting emphasis on parameter values with a high posterior probability. The posterior

apply it to sample the parameter space putting emphasis on parameter values with a high posterior probability. The posterior distribution is given in the Bayesian sense as the product of the likelihood and the prior distribution:

$$p(\boldsymbol{x}_{\text{mod}}|\boldsymbol{y}) \propto p(\boldsymbol{x}_{\text{mod}}|\boldsymbol{y},\boldsymbol{\beta})p(\boldsymbol{\beta})$$
 (7)

We denote the vector of the measured densities by y and the modelled densities by  $x_{mod}$  similar to Eqs. (1) and (3). To find the best parameters  $\beta$  for the model, we maximize  $\log p(x_{mod}|y)$ .

The likelihood  $p(\boldsymbol{x}_{mod}|\boldsymbol{y},\boldsymbol{\beta})$  is in our case given by a Gaussian distribution of the residuals, the difference of the model to the data, Eq. (8). Note that the normalization constant *C* in Eq. (8) does not influence the value of the maximal likelihood. The

$$p(\boldsymbol{x}_{\text{mod}}|\boldsymbol{y},\boldsymbol{\beta}) = \mathcal{N}(\boldsymbol{y},\mathbf{S}_y) = C \exp\left\{-\frac{1}{2}\left(\boldsymbol{y} - \boldsymbol{x}_{\text{mod}}(\boldsymbol{\beta})\right)^{\top} \mathbf{S}_y^{-1}\left(\boldsymbol{y} - \boldsymbol{x}_{\text{mod}}(\boldsymbol{\beta})\right)\right\}$$
(8)

The prior distribution  $p(\beta)$  restricts the parameters to lie within certain ranges, and the bounds we used for the sampling

covariance matrix  $\mathbf{S}_y$  contains the squared standard errors of the daily zonal means on the diagonal,  $\mathbf{S}_y = \text{diag}(\sigma_y^2)$ .

are listed in Table 1. Within those bounds we assume uniform (flat) prior distributions for the offset, the geomagnetic and solar amplitudes, and in the linear case also for the annual and semi-annual harmonics. We penalize large lifetimes using an exponential distribution  $p(\tau) \propto \exp\{-\tau/\sigma_{\tau}\}$  for each lifetime parameter, i.e. for  $\tau_0$ , d, and e in Eqs. (5) and (6). The scale width  $\sigma_{\tau}$  of this exponential distribution is fixed to one day. This choice of prior distributions for the lifetime parameters prevents sampling the edges of the parameter space at places with small geomagnetic coefficients. In those regions the lifetime may be ambiguous and less meaningful.

# 4.2 Correlations

In the simple case, the measurement covariance matrix  $\mathbf{S}_y$  contains the measurement uncertainties on the diagonal, in our case the (squared) standard error of the zonal means denoted by  $\sigma_y$ ,  $\mathbf{S}_y = \text{diag}(\sigma_y^2)$ . However, the standard error of the mean might underestimate the true uncertainties. In addition, possible correlations may occur which are not accounted for using a diagonal

$\mathbf{S}_y$ .

Both problems can be addressed by adding a covariance kernel **K** to  $S_y$ . Various forms of covariance kernels can be used (Rasmussen and Williams, 2006), depending on the underlying process leading to the measurement or residual uncertainties. Since we have no prior knowledge about the true correlations, we use a commonly chosen kernel of the Matérn-3/2 type (Matérn, 1960; MacKay, 2003; Rasmussen and Williams, 2006). This kernel depends only on the (time) distance between the measurements  $t_{ij} = |t_i - t_j|$  and has two parameters, the "strength"  $\sigma$  and correlation length  $\rho$ :

$$K_{ij} = \sigma^2 \left( 1 + \frac{\sqrt{3}t_{ij}}{\rho} \right) \exp\left\{ -\frac{\sqrt{3}t_{ij}}{\rho} \right\} \,. \tag{9}$$

Both parameters are estimated together with the model parameter vector  $\beta$ . We found that using the kernel (9) in a covariance matrix  $\mathbf{S}_{y}$  with the entries

$$S_{y_{ij}} = K_{ij} + \delta_{ij} \sigma_{y_i}^2 , \qquad (10)$$

15 worked best and led to stable and reliable parameter sampling. Note that an additional "white noise" term  $\sigma^2 \mathbb{1}$  could be added to the covariance matrix to account for still underestimated data uncertainties. However, this additional white noise term did not improve the convergence nor did it influence the fitted parameters significantly.

The approximately 3000x3000 covariance matrix of the Gaussian Process model for the residuals was evaluated using the Foreman-Mackey et al. (2017) approximation and the provided Python code (Foreman-Mackey et al., 2017). For onedimensional data sets, this approach is computationally faster than the full Cholesky decomposition which is usually used

to invert the covariance matrix S<sub>y</sub>. With this approximation, we achieved sensible Monte-Carlo sampling times to facilitate evaluating all 18x16 latitude x altitude bins on a small cluster in about one day. We used the emcee package (Foreman-Mackey et al., 2013) for the Monte-Carlo sampling, set up to use 112 walkers, 800 samples for the initial fit of the parameters, followed by another 800 so-called burn-in samples and 1400 production samples. The full code can be found at https://github.com/st-bender/sciapy.

# 5 Results

We demonstrate the parameter estimates using example time series  $x_{NO}$  at 70 km at 65°S, 5°N, and 65°N. NO shows different behaviour in these regions, showing the most variation with respect to the solar cycle and geomagnetic activity at high latitudes. In contrast, at low latitudes the geomagnetic influence should be reduced (Barth et al., 1988; Hendrickx et al., 2017; Kiviranta

30 et al., 2018). We briefly show only the results for the linear model and point out some of its shortcomings. Thereafter we show the results from the non-linear model and continue to use that for further analysis of the coefficients.