# Peer review of "Mesospheric nitric oxide model from SCIAMACHY data"

_Atmospheric Chemistry and Physics, 2018_

## Referee Comment (RC1) · Anonymous Referee #1 · 22 Nov 2018

This paper uses mesospheric NO observations taken by the SCIAMACHY instrument between 2002 and 2012 to build an non-linear empirical model of NO number densities as function of time and geomagnetic latitude, driven by the AE index as proxy for energetic particle precipitation and the Ly-a index as proxy for the EUV and XEUV influences.

The presented empirical model is very useful for constraining or validating atmospheric models and complements other empirical models of NO that focus either on the lower thermosphere or on the stratosphere/lower mesosphere.

The paper is generally well written and the methodology is presented in a clear manner. However, the discussion of the obtained responses could be improved by consideration of the limitations of the empirical model. For instance, modeled responses are

interpreted as NO production rates (e.g., the AE response) which is an oversimplification since important physical mechanisms are not represented by the model (e.g., transport). Similarly, many of the discussed hemispheric asymmetries are most likely related to the use of geomagnetic coordinates (see specific comments below). This should be addressed before publication in ACP.

Specific comments:

p2 l4: This is misleading: SSWs cause reduced mesospheric descent (or even an upwards motion), not enhanced descent. You are referring to the strong downwelling that often occurs in the recovery phase of the SSW, typically associated with the formation of an elevated stratopause.

p2 l9: MIPAS upper atmosphere observations were carried out in the 40 - 170 km range, see Bermejo-Pantaleón et al, 2011 (doi:10.1029/2011JA016752).

p2 l19: I wouldn't say the most CCMs parametrize NO. Rather, some models (those that are not resolving the thermosphere) constrain NO at the models upper lid by observation-based parameterizations.

p3 l5: Is there any atmospheric model study that illustrates the effectiveness of NOx photo-excitation as NO loss process? I would expect that N2O formation via metastable N2(A) +O2 as discussed in Funke et al. 2007 and Sheese et al., 2016 is likely a more relevant source for upper atmospheric N2O.

p3 l11-12: "NO is produced in dark conditions by particle precipitation at auroral latitudes, but is then depleted only by reacting with atomic nitrogen". I don't understand this sentence. NO is produced by particle precipitation at any illumination condition. NO loss is mostly occurring at sunlit conditions (photolysis of NO and subsequent reaction R5).

p3 l22: The semi-empirical model of Funke et al. does not only cover the stratosphere but also a significant part of the mesosphere.

p4 l22: "We use two proxies to model the NO number densities, one accounting for the long-term eleven-year solar cycle and one accounting for the short term geomagnetic activity." Isn't one of these proxies related to solar irradiance variations and the other one related to energetic particle precipitation? Both of them exhibit short- and longterm variability...

p5 l1: "questioned" seems too strong to me. Hendrickxs et al. simply noticed "that the auroral electrojet index is a more suitable proxy".

p5 l11: The choice of geomagnetic latitude as coordinate deserves some further discussion: Although production by EPP is linked to geomagnetic latitudes, mesospheric NO distributions are mostly ruled by illumination and (to some extend) by dynamics, both resulting in NO distributions organized in geographic latitudes.

p5 Eq3: What is the rationale behind omitting the harmonic term in the non-linear model? By doing so, seasonal variations not related to EPP remain unconsidered. Or, in other words, the model is forced to attribute any seasonal variation to EPP. In the same line, shouldn't a life-time correction be considered also for the Ly-a part?

p6 l7: "accounts for the different lifetime at polar night compared to polar day. " This could be phrased in a more general way, i.e., "...during winter and summer".

p10 l2: The vertical shape of the Ly-a response is intriguing: Why is this response peaking at 70-75 km while NO production due to XEUV should increase with altitude?

p11 l4: The larger amplitude of the NH annual lifetime variation is likely a result of the use of the geomagnetic latitude grid (this variation is smeared out in the SH due to the geomagnetic pole offset).

p12, l4-11: Similarly, the smaller SH AE coefficients (and shorter lifetimes) are likely related to the choice of a geomagnetic grid.

p13 l2: The AE coefficients do not represent a NO production rate, they simply represent the NO response to AE perturbations. Note that transport and mixing processes

are not considered by the empirical model, the latter being most likely responsible for the increased polar AE response around 70 km due to accumulation effects during the winter.

p14 l8: Note that the annually varying finite lifetime is only considered for the EPP-related part of NO.

––––––––––––––––––––––––––

---

## Referee Comment (RC2) · Anonymous Referee #2 · 19 Dec 2018

**General comments**

In this paper, Bender et al. describe the empirical model for NO that they have developed, based on SCIAMACHY limb measurements of mesospheric NO. This model relates the daily zonal means of NO number densities to the Lyman-$\alpha$ and AE indices, as proxies for the solar irradiance and geomagnetic activity, respectively.

Such a model is very useful, both to provide an estimate of the NO concentrations and to indicate regions where certain processes dominate. It is a good complement to other existing empirical models for NO in the MLT. These models can be used to validate or constrain atmospheric models, or as a tool to help compare different observational data sets with each other.

[Figure]

I recommend the publication of this article in ACP after consideration of the few revisions suggested below. My only significant comment is about the way the non-linear model is defined. The methodological choices could be described in a more accurate way. The comparison with the linear version of the model, as it is presented in the paper, is not convincing enough.

**Specific comments**

p.1, l.18-20: Variations in solar and geomagnetic activity do not affect the atmosphere only above 100 km. This is what one can understand when reading these few lines. Please reword this paragraph to make it clearer.

p.2, l.4-5: SSW events do not always result in a strong downward transport of air from the mesosphere to the upper stratosphere. The formation of an elevated stratopause is generally needed for that to be observed.

p.2, l.10: Odin/SMR NO measurements are actually available from 35 to 115 km (Pérot et al., 2014). As explained by Kiviranta et al. (2018), only a part of the available altitude range has been used to develop SANOMA model.

p.3, l.11: NO is produced by particle precipitation at auroral latitudes under sunlit conditions too.

p.3, l.14-15 and l.19: There is a mistake in the dates of the SNOE mission. According to Marsh et al. (2004), the instrument was operational for only two years, from 1998 to 2000.

p.4, l.22-23 and l.30: The Lyman-$\alpha$ index is a proxy for solar irradiance and the AE index is a proxy for geomagnetic activity, but both exhibit long- and short-term variations. Please reword your description of the proxies.

p.4-5, Sec. 2.2: Please indicate the source for the proxy data used in your study.

p.5, l.1: I agree with referee 1 about the fact that "questioned" seems too strong. Hendrickx et al. (I guess that you meant 2015, and not 2017) showed that the AE index correlates better with NO concentrations measured by SOFIE. One cannot draw a conclusion from one study based on one instrument. As mentioned in Kiviranta et al. (2018), the Kp index correlates better than the AE index with Odin/SMR NO observations.

p.5, l.25: "We then omit the harmonic parts in the model." Why? Please explain this choice.

p.9, l.10-11: "At high southern and low latitudes, the improvement over the linear model is less evident." How do you explain that the improvement is clearer in the NH than in the SH?

Fig. 1-2 - following my previous comment: For high southern and low latitudes, it is difficult to see any clear difference between the linear and non-linear models in the residuals but, looking at the upper panels, it actually seems that the linear model reproduce better the seasonal variations of the data. That would mean that the non-linear model is not better in all regions. That could be due to the fact that, in your non-linear version of the model, you do not take into account seasonal variations which are not related to EPP. Please comment about that.

p.10, l.1-2: How do you explain the observed decrease in the Lyman-$\alpha$ parameter distribution between 80 and 90 km?

p.10, l.2-3: "The Lyman-$\alpha$ coefficients are all negative below 65 km." Please mention that the coefficients are negative at high northern latitudes too.

p.11, l.5: "The amplitude also increases with decreasing altitude." This is not always true. At high northern latitudes, the amplitude actually decreases with decreasing altitude between 75 and 90 km. Please make the description of this figure more accurate and comment about this observed distribution (in Sec. 5.4).

p.12, l.16-17: "We observe negative Lyman-$\alpha$ coefficients [...] at high northern latitudes above 80 km." How do you explain that such a pattern is observed only in the NH?

p.13, l.2: "Production rate" I agree with referee 1 on this point. AE coefficients do not represent the NO production rate, but rather the NO response to changes in the AE index.

p.13, l.17-18: "the increasing photochemical lifetime at low solar zenith angles." This sentence is unclear. According to Sinnhuber et al. (2016, Fig.7b), the photochemical lifetime of NO is lower for low solar zenith angles than for high SZAs. Did you mean "the increasing photochemical lifetime with decreasing altitude, at low SZAs"? In any case, this does not explain why the annual variation of your lifetime parameter increases from 75 to 90 km in the highest northern latitude bin.

**Technical corrections**

Fig.1 and 2: These figures are not easy to read, especially because the dots representing the data are hidden by the error bars. Maybe you could represent the error bars in a different way (other colour for example) in order to make the plots clearer.

p.9, l.17-19: This sentence is unclear (too long). Please reword.

p.18, l.1: "Versick, S" has been written twice.

---

## Author Response (AR1)

We thank the reviewer for the careful review of the manuscript. We have marked our replies in blue and changes made in the paper in green .

**Anonymous Referee #1**

This paper uses mesospheric NO observations taken by the SCIAMACHY instrument between 2002 and 2012 to build an non-linear empirical model of NO number densities as function of time and geomagnetic latitude, driven by the AE index as proxy for energetic particle precipitation and the Ly-a index as proxy for the EUV and XEUV influences.

The presented empirical model is very useful for constraining or validating atmospheric models and complements other empirical models of NO that focus either on the lower thermosphere or on the stratosphere/lower mesosphere.

The paper is generally well written and the methodology is presented in a clear manner. However, the discussion of the obtained responses could be improved by consideration of the limitations of the empirical model. For instance, modeled responses are interpreted as NO production rates (e.g., the AE response) which is an oversimplification since important physical mechanisms are not represented by the model (e.g., transport). Similarly, many of the discussed hemispheric asymmetries are most likely related to the use of geomagnetic coordinates (see specific comments below). This should be addressed before publication in ACP.

We address the issues raised in our replies to the specific comments below.

Specific comments:

p2 l4: This is misleading: SSWs cause reduced mesospheric descent (or even an upwards motion), not enhanced descent. You are referring to the strong downwelling that often occurs in the recovery phase of the SSW, typically associated with the formation of an elevated stratopause.
We thank the reviewer for the pointer and changed the sentence accordingly:
"Additional dynamical processes also result in a strong downward transport of mesospheric air into the upper stratosphere, such as the strong downwelling that often occurs in the recovery phase of a Sudden Stratospheric Warming (SSW) (Orsolini et al., 2017; Pérot et al., 2014). This downwelling is typically associated with the formation of an elevated stratopause."

p2 l9: MIPAS upper atmosphere observations were carried out in the 40 - 170 km range, see Bermejo-Pantaleón et al, 2011 (doi:10.1029/2011JA016752).

Indeed, Bermejo-Pantaleón et al., 2011 states an altitude range of 42–172 km for the MIPAS UA observations. We changed the numbers in the text and added (Bermejo-Pantaleón et al., 2011) to the citations.

p2 l19: I wouldn't say the most CCMs parametrize NO. Rather, some models (those that are not resolving the thermosphere) constrain NO at the models upper lid by observation-based parameterizations.

We agree with the reviewer and to clarify, we changed the sentence in question to read:

"…some models constrain the NO content at their top layer by observation-based parameterizations."

We also added a reference to (Funke et al., 2016) after "…Envisat/MIPAS NO measurements" in the next sentence since this parameterization is suggested in (Matthes et al., 2017).

p3 l5: Is there any atmospheric model study that illustrates the effectiveness of NOx photo-excitation as NO loss process? I would expect that N2O formation via metastable N2(A) +O2 as discussed in Funke et al. 2007 and Sheese et al., 2016 is likely a more relevant source for upper atmospheric N2O.

We agree with the reviewer that a thorough study has yet to be carried out. To explain the possible connection between $N_2O$ and our study, we have changed and extended the discussion after reaction (R5) to:

"$N_2O$ has been retrieved in the mesosphere and thermosphere from MIPAS (see, e.g. Funke et al. (2008b), Funke et al. (2008a)) and from Scisat-1/ACE-FTS (Sheese et al., 2016). Model–measurement studies by Semeniuk et al. (2008) attributed the source of this $N_2O$ to being most likely the reaction between $NO_2$ and N atoms produced by particle precipitation:

$$N + NO_2 \rightarrow N_2O + O \ . \tag{R6}$$

We note that photo-excitation and photolysis at 185 nm (vacuum UV) of NO or $NO_2$ mixtures in nitrogen, $N_2$, or helium mixtures at 1 atm leads to $N_2O$ formation (Maric and Burrows, 1992). Both mechanisms explaining the production of $N_2O$ involve excited states of NO. Hence these pathways contribute to the loss of NO and potentially an additional daytime source of $N_2O$ in the upper atmosphere. $N_2O$ acts as an intermediate reservoir at high altitudes …"

p3 l11-12: "NO is produced in dark conditions by particle precipitation at auroral latitudes, but is then depleted only by reacting with atomic nitrogen". I don't understand this sentence. NO is produced by particle precipitation at any illumination condition. NO loss is mostly occurring at sunlit conditions (photolysis of NO and subsequent reaction R5).

We agree that in the way it is expressed, it is indeed misleading. We reordered this sentence to read:

"NO is produced by particle precipitation at auroral latitudes, but in dark conditions (without photolysis) it is depleted only by reacting with atomic nitrogen (reaction (R5))."

p3 l22: The semi-empirical model of Funke et al. does not only cover the stratosphere but also a significant part of the mesosphere.

We added: "and mesosphere" after "stratosphere" to that sentence.

p4 l22: "We use two proxies to model the NO number densities, one accounting for the long-term eleven-year solar cycle and one accounting for the short term geomagnetic activity." Isn't one of these proxies related to solar irradiance variations and the other one related to energetic particle precipitation? Both of them exhibit short- and longterm variability. . .

Our expression was indeed misleading and did not convey the meaning we intended (which is more accurately described by the reviewer's words). Thus, we changed the beginning of the paragraph to read:
"We use two proxies to model the NO number densities, one accounting for the solar irradiance variations and one accounting for the geomagnetic activity. Various proxies have been used or proposed to account for the solar irradiance induced variations in mesospheric–thermospheric NO, which are in particular related to the eleven-year solar cycle."

p5 l1: "questioned" seems too strong to me. Hendrickxs et al. simply noticed "that the auroral electrojet index is a more suitable proxy".
We rephrased the sentence to read:
"However, Hendrickx et al. (2015) found that the auroral electrojet index (AE) (Davis and Sugiura, 1966) correlated better with SOFIE-derived NO concentrations (Hendrickx et al., 2015, 2017) (see also Sinnhuber et al., 2016)."
We further changed "matter of debate" to "matter of opinion" at the beginning of the paragraph in question.

p5 l11: The choice of geomagnetic latitude as coordinate deserves some further discussion: Although production by EPP is linked to geomagnetic latitudes, mesospheric NO distributions are mostly ruled by illumination and (to some extend) by dynamics, both resulting in NO distributions organized in geographic latitudes.
Geomagnetic latitudes were used by (Marsh et al., 2004), (Sinnhuber et al., 2016), and (Kiviranta et al., 2018), because particle-induced NO production is related to geomagnetic latitudes. We added the following note to Sect. 2.1:
"Note that mesospheric NO concentrations are related to geomagnetically as well as geographically based processes, but disentangling them is beyond the scope of the paper. Follow-up studies can build on the method presented here and study, for example, longitudinally resolved timeseries."
We then added a reference in the mentioned place.
"… function of the (geomagnetic, see Sect. 2.1) latitude $\phi$, …

p5 Eq3: What is the rationale behind omitting the harmonic term in the non-linear model? By doing so, seasonal variations not related to EPP remain unconsidered. Or, in other words, the model is forced to attribute any seasonal variation to EPP. In the same line, shouldn't a life-time correction be considered also for the Ly-a part?
When there is strong UV-induced NO production, it is naturally depleted by photolysis (R3) or photoionization (R4) at the same time. The lifetime is therefore expected to be < 1 day. We tested a finite lifetime for the Lyman-$\alpha$ part, but found that this only increases the number of parameters without improving the fit quality.

We found that we could achieve similar or better fits to the data already without the harmonic terms in the non-linear model. We added the following text after Eq. (3):
"Although this approach shifts all seasonal variations to the AE index and thus attributes them to particle-induced effects, we found that the residual traces of particle-unrelated seasonal effects were minor compared to the overall improvement of the fit. Additional harmonic terms increase only the number of free parameters without substantially improving the fit further."

p6 l7: "accounts for the different lifetime at polar night compared to polar day. " This could be phrased in a more general way, i.e., ". . .during winter and summer".
We agree with the reviewer and changed the sentence to end in the suggested way.

p10 l2: The vertical shape of the Ly-a response is intriguing: Why is this response peaking at 70-75 km while NO production due to XEUV should increase with altitude?

The Lyman-$\alpha$ index predominantly describes the 11-year solar cycle. At middle and low latitudes, the time-series themselves show a pronounced solar cycle variation at this altitude range, and they get "flatter" above 75 km until the solar cycle is visible again at around 88 and 90 km. We added the following note after that sentence:

"The penetration of Lyman-$\alpha$ radiation decreases with decreasing altitude as a result of scattering and absorption by air molecules. On the other hand the concentration of air decreases with altitude. At this stage we have not an unambiguous explanation of this behaviour, but it may be related to reaction pathways as laid out by (Pendleton et al., 1983) which would relate the NO concentrations to the $CO_2$ and $H_2O$ (or OH, respectively) profiles."

On a related issue we discovered that the stated $N_2$ dissociation energy is wrong. We therefore changed the statement below (R2) to:

"The dissociation energy of $N_2$ into ground state atoms $N(^4S)$ is about 9.8 eV ($\lambda \approx 127$ nm) (Frost et al., 1956; Heays et al., 2017; Hendrie, 1954). This energy together with the excitation energy to $N(^2D)$ is denoted by $h\nu$ in (R1) and can be provided by a number of sources, most notably by auroral or photoelectrons as well as by soft solar X-rays."

And we added "($\lambda < 102$ nm)" to reaction (R1).

p11 l4: The larger amplitude of the NH annual lifetime variation is likely a result of the use of the geomagnetic latitude grid (this variation is smeared out in the SH due to the geomagnetic pole offset).

We agree with the reviewer and added the following statement:

"This difference could be linked to the geomagnetic latitudes which include a wider range of geographic latitudes in the Southern Hemisphere compared to the Northern Hemisphere. Therefore, the annual variation is less apparent in the Southern Hemisphere."

p12, l4-11: Similarly, the smaller SH AE coefficients (and shorter lifetimes) are likely related to the choice of a geomagnetic grid.

This is discussed at the end of that paragraph and we added another possible explanation:

"A third possibility may be the exclusion of the Southern Atlantic Anomaly from the retrieval (Bender et al., 2013, 2017) where presumably the particle-induced impact on NO is largest."

p13 l2: The AE coefficients do not represent a NO production rate, they simply represent the NO response to AE perturbations. Note that transport and mixing processes are not considered by the empirical model, the latter being most likely responsible for the increased polar AE response around 70 km due to accumulation effects during the winter.

It is true that the AE coefficients are not production rates, we added the following statement to explain the conversion better:

"The AE coefficient can be considered as an effective production rate modulated by all short-time ($\ll$ 1 day) processes. To roughly estimate this production rate, we divided the coefficient of the (daily) AE by 86400 s which follows the approach in (Sinnhuber et al., 2016). We find a maximum production rate of about $1 \text{ cm}^{-3} \text{ nT}^{-1} \text{ s}^{-1}$ around 70–72 km."

We also changed the Figs. 3, 5, and 6 to show the original coefficients instead of the converted "production rates". We already include an accumulation effect by our finite lifetime approach, see Eq. (4).

p14 l8: Note that the annually varying finite lifetime is only considered for the EPP-related part of NO.

We changed the sentence to end:

"… using an annually varying finite lifetime for the particle-induced NO."
* * *
Bender, S., Sinnhuber, M., Burrows, J. P., Langowski, M., Funke, B. and López-Puertas, M.: Retrieval of nitric oxide in the mesosphere and lower thermosphere from SCIAMACHY limb spectra, Atmos. Meas. Tech., 6(9), 2521–2531, doi:10.5194/amt-6-2521-2013, 2013.

Bender, S., Sinnhuber, M., Langowski, M. and Burrows, J. P.: Retrieval of nitric oxide in the mesosphere from SCIAMACHY nominal limb spectra, Atmos. Meas. Tech., 10(1), 209–220, doi:10.5194/amt-10-209-2017, 2017.

Bermejo-Pantaleón, D., Funke, B., López-Puertas, M., García-Comas, M., Stiller, G. P., Clarmann, T. von, Linden, A., Grabowski, U., Höpfner, M., Kiefer, M., Glatthor, N., Kellmann, S. and Lu, G.: Global observations of thermospheric temperature and nitric oxide from mipas spectra at 5.3 m, J. Geophys. Res., 116(A10), A10313, doi:10.1029/2011JA016752, 2011.

Davis, T. N. and Sugiura, M.: Auroral electrojet activity index AE and its universal time variations, J. Geophys. Res., 71(3), 785–801, doi:10.1029/jz071i003p00785, 1966.

Frost, D. C., McDowell, C. A. and Bawn, C. E. H.: The dissociation energy of the nitrogen molecule, Proceedings of the Royal Society of London. Series A. Mathematical and Physical Sciences, 236(1205), 278–284, doi:10.1098/rspa.1956.0135, 1956.

Funke, B., Garcı'a-Comas, M., López-Puertas, M., Glatthor, N., Stiller, G. P., Clarmann, T. von, Semeniuk, K. and McConnell, J. C.: Enhancement of $N_2O$ during the octoberNovember 2003 solar proton events, Atmos. Chem. Phys., 8(14), 3805–3815, doi:10.5194/acp-8-3805-2008, 2008a.

Funke, B., López-Puertas, M., Garcia-Comas, M., Stiller, G. P., Clarmann, T. von and Glatthor, N.: Mesospheric $N_2O$ enhancements as observed by MIPAS on envisat during the polar winters in 20022004, Atmos. Chem. Phys., 8(19), 5787–5800, doi:10.5194/acp-8-5787-2008, 2008b.

Funke, B., López-Puertas, M., Stiller, G. P., Versick, S. and Clarmann, T. von: A semi-empirical model for mesospheric and stratospheric $NO_y$ produced by energetic particle precipitation, Atmos. Chem. Phys., 16, 8667–8693, doi:10.5194/acp-16-8667-2016, 2016.

Heays, A. N., Bosman, A. D. and Dishoeck, E. F. van: Photodissociation and photoionisation of atoms and molecules of astrophysical interest, Astronomy & Astrophysics, 602, A105, doi:10.1051/0004-6361/201628742, 2017.

Hendrickx, K., Megner, L., Gumbel, J., Siskind, D. E., Orsolini, Y. J., Tyssøy, H. N. and Hervig, M.: Observation of 27 day solar cycles in the production and mesospheric descent of EPP-produced NO, J. Geophys. Res., 120(10), 8978–8988, doi:10.1002/2015JA021441, 2015.

Hendrickx, K., Megner, L., Marsh, D. R., Gumbel, J., Strandberg, R. and Martinsson, F.: Relative Importance of Nitric Oxide Physical Drivers in the Lower Thermosphere, Geophys. Res. Lett., 44(19), 10, 081–10, 087, doi:10.1002/2017gl074786, 2017.

Hendrie, J. M.: Dissociation energy of $N_2$, The Journal of Chemical Physics, 22(9), 1503–1507, doi:10.1063/1.1740449, 1954.

Kiviranta, J., Pérot, K., Eriksson, P. and Murtagh, D.: An empirical model of nitric oxide in the upper mesosphere and lower thermosphere based on 12 years of Odin SMR measurements, Atmos. Chem. Phys., 18(18), 13393–13410, doi:10.5194/acp-18-13393-2018, 2018.

Maric, D. and Burrows, J.: Formation of $N_2O$ in the photolysis/photoexcitation of NO, $NO_2$ and air, J. Photochem. Photobiol., A, 66(3), 291–312, doi:10.1016/1010-6030(92)80002-d, 1992.

Marsh, D. R., Solomon, S. C. and Reynolds, A. E.: Empirical model of nitric oxide in the lower thermosphere, J. Geophys. Res., 109(A7), A07301, doi:10.1029/2003JA010199, 2004.

Matthes, K., Funke, B., Andersson, M. E., Barnard, L., Beer, J., Charbonneau, P., Clilverd, M. A., Wit, T. D. de, Haberreiter, M., Hendry, A., Jackman, C. H., Kretzschmar, M., Kruschke, T., Kunze, M., Langematz, U., Marsh, D. R., Maycock, A. C., Misios, S., Rodger, C. J., Scaife, A. A., Seppälä, A., Shangguan, M., Sinnhuber, M., Tourpali, K., Usoskin, I., Kamp, M. van de, Verronen, P. T. and Versick, S.: Solar forcing for CMIP6 (v3.2), Geoscientific Model Development, 10, 2247–2302, doi:10.5194/gmd-10-2247-2017, 2017.

Orsolini, Y. J., Limpasuvan, V., Pérot, K., Espy, P., Hibbins, R., Lossow, S., Raaholt Larsson, K. and Murtagh, D.: Modelling the descent of nitric oxide during the elevated stratopause event of january 2013, J. Atmos. Sol. Terr. Phys., 155, 50–61, doi:10.1016/j.jastp.2017.01.006, 2017.

Pendleton, W., Erman, P., Larsson, M. and Witt, G.: Observation of strong NO gamma-band radiation induced in thin n2-CO2and n2-h2o targets by electron impact and its possible relation to the auroral chemistry of NO, Physica Scripta, 28(5), 532–538, doi:10.1088/0031-8949/28/5/005, 1983.

Pérot, K., Urban, J. and Murtagh, D. P.: Unusually strong nitric oxide descent in the Arctic middle atmosphere in early 2013 as observed by Odin/SMR, Atmos. Chem. Phys., 14(15), 8009–8015, doi:10.5194/acp-14-8009-2014, 2014.

Semeniuk, K., McConnell, J. C., Jin, J. J., Jarosz, J. R., Boone, C. D. and Bernath, P. F.: $N_2O$ production by high energy auroral electron precipitation, J. Geophys. Res., 113(D16), doi:10.1029/2007jd009690, 2008.

Sheese, P. E., Walker, K. A., Boone, C. D., Bernath, P. F. and Funke, B.: Nitrous oxide in the atmosphere: First measurements of a lower thermospheric source, Geophys. Res. Lett., 43(6), 2866–2872, doi:10.1002/2015gl067353, 2016.

Sinnhuber, M., Friederich, F., Bender, S. and Burrows, J. P.: The response of mesospheric NO to geomagnetic forcing in 2002–2012 as seen by SCIAMACHY, J. Geophys. Res., 121(4), 3603–3620, doi:10.1002/2015JA022284, 2016.

Atmos. Chem. Phys. Discuss.,

https://doi.org/10.5194/acp-2018-872-RC2, 2018

We thank the reviewer for the careful review of the manuscript. We have marked our replies in blue and changes made in the paper in green .

**Anonymous Referee #2**

**General comments**

In this paper, Bender et al. describe the empirical model for NO that they have developed, based on SCIA-MACHY limb measurements of mesospheric NO. This model relates the daily zonal means of NO number densities to the Lyman-$\alpha$ and AE indices, as proxies for the solar irradiance and geomagnetic activity, respectively.

Such a model is very useful, both to provide an estimate of the NO concentrations and to indicate regions where certain processes dominate. It is a good complement to other existing empirical models for NO in the MLT. These models can be used to validate or constrain atmospheric models, or as a tool to help compare different observational data sets with each other.

I recommend the publication of this article in ACP after consideration of the few revisions suggested below. My only significant comment is about the way the non-linear model is defined. The methodological choices could be described in a more accurate way. The comparison with the linear version of the model, as it is presented in the paper, is not convincing enough.

We address the issues raised in our replies to the specific comments below.

**Specific comments**

p.1, l.18-20: Variations in solar and geomagnetic activity do not affect the atmosphere only above 100 km. This is what one can understand when reading these few lines. Please reword this paragraph to make it clearer.
We changed the sentence to:
"The atmosphere above the stratosphere ($\gtrsim$ 40 km) is coupled to solar and ..."

p.2, l.4-5: SSW events do not always result in a strong downward transport of air from the mesosphere to the upper stratosphere. The formation of an elevated stratopause is generally needed for that to be observed.
We refer to our reply to reviewer #1 on the same issue

p.2, l.10: Odin/SMR NO measurements are actually available from 35 to 115 km (Pérot et al., 2014). As explained by Kiviranta et al. (2018), only a part of the available altitude range has been used to develop SANOMA model.
The altitude range mentioned is not stated in that reference, although the figures presented therein cover a pressure range from 9 hPa ($\approx$ 32 km) to 0.003 hPa ($\approx$ 90 km). The level 2 data available on http://odin.rss.chalmers.se/level2 (referred to by Kiviranta et al., 2018) cover the pressure range from 1 hPa ($\approx$ 48 km) to 1e-5 hPa ($\approx$ 133 km). There the official denoted altitude range is 45 to 115 km and we changed the numbers in the text to that altitude range.

p.3, l.11: NO is produced by particle precipitation at auroral latitudes under sunlit conditions too.
We agree that the used expression is misleading, the text has been changed, see reply to reviewer #1 on the same point.

p.3, l.14-15 and l.19: There is a mistake in the dates of the SNOE mission. According to Marsh et al. (2004), the instrument was operational for only two years, from 1998 to 2000.
The SNOE mission lasted longer, until December 2003 (Bailey, 2005), but only part of the data have been used by (Marsh et al., 2004) to develop the NOEM model. We corrected the SNOE data period on which NOEM is based to 03/1998–09/2000, and we changed. "three years" to "two and a half years".

p.4, l.22-23 and l.30: The Lyman-α index is a proxy for solar irradiance and the AE index is a proxy for geomagnetic activity, but both exhibit long- and short-term variations. Please reword your description of the proxies.
We changed the beginning of the section according to our reply to the same point raised by reviewer #1. In addition we changed the beginning of the second paragraph to read:
"In the same manner as for the irradiance variations, the"right" geomagnetic index to model particle-induced variations of NO is a matter of opinion."

p.4-5, Sec. 2.2: Please indicate the source for the proxy data used in your study.
We added the sources for the proxy data to the "Code and data availability" paragraph:
"The solar Lyman-$\alpha$ index data were downloaded from `http://lasp.colorado.edu/lisird/data/composite_lyman_alpha/`, the AE index data were downloaded from `http://wdc.kugi.kyoto-u.ac.jp/aedir/`, and the daily mean values used in this study are available within the aforementioned data set."

p.5, l.1: I agree with referee 1 about the fact that "questioned" seems too strong. Hendrickx et al. (I guess that you meant 2015, and not 2017) showed that the AE index correlates better with NO concentrations measured by SOFIE. One cannot draw a conclusion from one study based on one instrument. As mentioned in Kiviranta et al. (2018), the Kp index correlates better than the AE index with Odin/SMR NO observations.
Our work relates more closely to (Hendrickx et al., 2017), but indeed, the motivation to use AE for SOFIE NO data is discussed in more detail in (Hendrickx et al., 2015). We changed the text to read according to our reply to the same point raised by reviewer #1, also changing the reference to (Hendrickx et al., 2015). The (Kiviranta et al., 2018) study uses Odin/SMR NO above 85 km, where Kp may be better suited. In our case AE resulted in better fits and Kp may not be really suited because of its non-linearity. The effect

of using other indices is discussed in the original manuscript in the paragraph below the mentioned place. However, in addition to the aforementioned change, we changed "Kp" to "Kp (or its linear equivalent Ap)" in the following paragraph.

p.5, l.25: "We then omit the harmonic parts in the model." Why? Please explain this choice.
We refer to our reply to reviewer #1 on the same issue.

p.9, l.10-11: "At high southern and low latitudes, the improvement over the linear model is less evident." How do you explain that the improvement is clearer in the NH than in the SH?
We added the following explanation after the mentioned sentence:
"At low latitudes, the NO content is apparently mostly related to the eleven-year solar cycle and the particle influence is suppressed. Since this cycle is covered by the Lyman-$\alpha$ index, both models perform similarly, but the non-linear version has one less parameter. At high southern latitudes, the SCIAMACHY data are less densely sampled compared to high northern latitudes (see (Bender et al., 2017)). In addition to the sampling differences, geomagnetic latitudes encompass a wider geographic range in the Southern Hemisphere (SH) than in the Northern Hemisphere (NH), and the AE index is derived from stations in the Northern Hemisphere. Both effects can lower the NO concentrations that SCIAMACHY observes in the Southern Hemisphere particularly at the winter maxima. The lifetime variation that improves the fit in the NH is thus less effective in the SH."

Fig. 1-2 - following my previous comment: For high southern and low latitudes, it is difficult to see any clear difference between the linear and non-linear models in the residuals but, looking at the upper panels, it actually seems that the linear model reproduce better the seasonal variations of the data. That would mean that the non-linear model is not better in all regions. That could be due to the fact that, in your non-linear version of the model, you do not take into account seasonal variations which are not related to EPP. Please comment about that.
One should be careful when comparing the upper panels in Figs. 1 and 2 because the blue line can be misleading. We therefore presented the residuals on which the judgement should be based. The mentioned apparent traces of residual seasonal effects in the non-linear fit are minor compared to the overall lower residuals, keeping in mind that the non-linear model has one parameter less than the linear version. We added the following note to the discussion of the non-linear fit quality (including the footnote):
"In both regions the residuals show traces of seasonal variations that are not related to particle effects. The linear model appears to capture these variations better than the non-linear model. However, by objective measures including the number of model parameters,[1] the non-linear version fits the data better in all bins (not shown here)."

p.10, l.1-2: How do you explain the observed decrease in the Lyman-α parameter distribution between 80 and 90 km?
We refer to our reply to reviewer #1 on the same issue where we extended the discussion about the Lyman-$\alpha$
* * *
[1] Past and recent research in model selection provides a number of choices on how to compare models objectively. The results are so-called information criteria which aim to provide a consistent way of how to compare models, most notably the "Akaike Information Criterion" (AIC, (Akaike, 1974)), the "Bayesian Information Criterion" or "Schwarz Criterion" (BIC or SIC, (Schwarz, 1978)), the "Deviance Information Criterion" (DIC, (Ando, 2011; Spiegelhalter et al., 2002)), or the "Widely Applicable Information Criterion" (WAIC, (Vehtari et al., 2016; Watanabe, 2010)). Alternatively, the "Standardized Mean Squared Error" (SMSE) or the "Mean Standardized Log-Loss" (MSLL) (Rasmussen and Williams, 2006, Ch. 2) give an impression of the quality of regression models with respect to each other.

parameter morphology.

p.10, l.2-3: "The Lyman-α coefficients are all negative below 65 km." Please mention that the coefficients are negative at high northern latitudes too.
We added the following statement and short discussion after the mentioned sentence:
"We also observe negative values at high northern latitude at all altitudes and at high southern latitudes above 85 km. These negative coefficients indicate that NO photodissociation or conversion to other species outweighs its production via UV radiation in those places. The north–south asymmetry may be related to sampling and the difference in illumination with respect to geomagnetic latitudes, see Sect. 5.1."

p.11, l.5: "The amplitude also increases with decreasing altitude." This is not always true. At high northern latitudes, the amplitude actually decreases with decreasing altitude between 75 and 90 km. Please make the description of this figure more accurate and comment about this observed distribution (in Sec. 5.4).
The results for the highest northern latitude bin should be taken with caution. We changed the line in question:
"The amplitude also increases with decreasing altitude below 75 km at middle and high latitudes and with increasing altitude above that. The increasing annual variation at low altitudes can be the result of transport processes …"
We then added the following statement to the discussion (Sect. 5.4):
"Note that the results (in particular the large annual variation) in the northernmost latitude bin should be taken with caution because this bin is sparsely sampled by SCIAMACHY and the large winter NO concentrations are actually absent from the data."

p.12, l.16-17: "We observe negative Lyman-α coefficients […] at high northern latitudes above 80 km." How do you explain that such a pattern is observed only in the NH?
This remark is related to the point raised above referring to p.10, l.2-3 and the explanation is similar. We added the following text to the end of that paragraph:
"At high southern latitudes these negative Lyman-$\alpha$ coefficients are not as pronounced as at high northern latitudes. As mentioned in Sect. 5.2, this north–south asymmetry may be related to sampling and the difference in illumination with respect to geomagnetic latitudes, see also Sect. 5.1."

p.13, l.2: "Production rate" I agree with referee 1 on this point. AE coefficients do not represent the NO production rate, but rather the NO response to changes in the AE index.
We refer to our reply to reviewer #1 on the same issue.

p.13, l.17-18: "the increasing photochemical lifetime at low solar zenith angles." This sentence is unclear. According to Sinnhuber et al. (2016, Fig.7b), the photochemical lifetime of NO is lower for low solar zenith angles than for high SZAs. Did you mean "the increasing photochemical lifetime with decreasing altitude, at low SZAs"? In any case, this does not explain why the annual variation of your lifetime parameter increases from 75 to 90 km in the highest northern latitude bin.
The "low solar zenith angles" is a typo, we changed "low" to "large".
The issue with the northernmost latitude bin is addressed in our reply above (p.11, l.5).

**Technical corrections**

Fig.1 and 2: These figures are not easy to read, especially because the dots representing the data are hidden by the error bars. Maybe you could represent the error bars in a different way (other colour for example) in order to make the plots clearer.

We thinned the error bars and changed their colour to gray. However, due to the high density of data points, this approach helps only a little, but we hope that the quality improved so that the data variations are visible better.

p.9, l.17-19: This sentence is unclear (too long). Please reword.

We changed the last two sentences in that paragraph to read:

"Since we require the geomagnetic index and constant lifetime parameters to be larger than zero (see Table 1), these sampled distributions are sometimes skewed towards zero even though the 95% credible region is still larger than zero. Excluding heavily skewed distributions avoids those cases because the "true" parameter is apparently zero."

p.18, l.1: "Versick, S" has been written twice.

Fixed.
* * *
Akaike, H.: A new look at the statistical model identification, IEEE Transactions on Automatic Control, 19(6), 716–723, doi:10.1109/tac.1974.1100705, 1974.

Ando, T.: Predictive bayesian model selection, American Journal of Mathematical and Management Sciences, 31(1-2), 13–38, doi:10.1080/01966324.2011.10737798, 2011.

Bailey, S. M.: Observations of polar mesospheric clouds by the Student Nitric Oxide Explorer, J. Geophys. Res., 110(D13), D13203, doi:10.1029/2004jd005422, 2005.

Bender, S., Sinnhuber, M., Langowski, M. and Burrows, J. P.: Retrieval of nitric oxide in the mesosphere from SCIAMACHY nominal limb spectra, Atmos. Meas. Tech., 10(1), 209–220, doi:10.5194/amt-10-209-2017, 2017.

Hendrickx, K., Megner, L., Gumbel, J., Siskind, D. E., Orsolini, Y. J., Tyssøy, H. N. and Hervig, M.: Observation of 27 day solar cycles in the production and mesospheric descent of EPP-produced NO, J. Geophys. Res., 120(10), 8978–8988, doi:10.1002/2015JA021441, 2015.

Hendrickx, K., Megner, L., Marsh, D. R., Gumbel, J., Strandberg, R. and Martinsson, F.: Relative Importance of Nitric Oxide Physical Drivers in the Lower Thermosphere, Geophys. Res. Lett., 44(19), 10, 081–10, 087, doi:10.1002/2017gl074786, 2017.

Kiviranta, J., Pérot, K., Eriksson, P. and Murtagh, D.: An empirical model of nitric oxide in the upper mesosphere and lower thermosphere based on 12 years of Odin SMR measurements, Atmos. Chem. Phys., 18(18), 13393–13410, doi:10.5194/acp-18-13393-2018, 2018.

Marsh, D. R., Solomon, S. C. and Reynolds, A. E.: Empirical model of nitric oxide in the lower thermosphere, J. Geophys. Res., 109(A7), A07301, doi:10.1029/2003JA010199, 2004.

Rasmussen, C. E. and Williams, C. K. I.: Gaussian processes for machine learning, MIT Press, Cambridge, MA. [online] Available from: http://www.gaussianprocess.org/gpml/chapters/, 2006.

Schwarz, G.: Estimating the dimension of a model, The Annals of Statistics, 6(2), 461–464, doi:10.1214/aos/1176344136, 1978.

Spiegelhalter, D. J., Best, N. G., Carlin, B. P. and Linde, A. van der: Bayesian measures of model complexity and fit, Journal of the Royal Statistical Society: Series B (Statistical Methodology), 64(4), 583–639, doi:10.1111/1467-9868.00353, 2002.

Vehtari, A., Gelman, A. and Gabry, J.: Practical bayesian model evaluation using leave-one-out cross-validation and WAIC, Statistics and Computing, 27(5), 1413–1432, doi:10.1007/s11222-016-9696-4, 2016.

Watanabe, S.: Asymptotic equivalence of Bayes cross validation and widely applicable information criterion in singular learning theory, Journal of Machine Learning Research (JMLR), 11, 3571–3594 [online] Available from: http://www.jmlr.org/papers/v11/watanabe10a.html, 2010.

**Mesospheric nitric oxide model from SCIAMACHY data**

Stefan Bender[1], Miriam Sinnhuber[2], Patrick J. Espy[1], and John P. Burrows[3]

[1]Norwegian University of Science and Technology, Trondheim, Norway
[2]Karlsruhe Institute of Technology, Karlsruhe, Germany
[3]University of Bremen, Bremen, Germany

**Correspondence:** Stefan Bender (stefan.bender@ntnu.no)

[revised manuscript text omitted]

25   derived partly from Envisat/MIPAS NO measurements (Funke et al., 2016).

**NO in the Mesosphere and Lower Thermosphere**

NO in the mesosphere and lower thermosphere is produced by $N_2$ dissociation

$$N_2 + h\nu \rightarrow \underline{N(^2D)}N(^2D) + \underline{N}N(^4S) \quad (\underline{^4S}\lambda < 102\,nm) , \tag{R1}$$

followed by the reaction of the excited nitrogen atom  N(²D) with molecular oxygen (Solomon et al., 1982; Barth, 1992,

30   1995):

$$\underline{N(^2D)}N(^2D) + O_2 \rightarrow NO + O . \tag{R2}$$

The  dissociation energy of $N_2$  into ground state atoms $N(^4S)$ is about 9.8 eV  ($\lambda \approx 127$ nm) (Hendrie, 1954; Frost et al., 1956; Heays et al., 2017). This energy together with the excitation energy to $N(^2D)$ is denoted by $h\nu$ in (R1)  and can be provided by a number of sources, most notably by auroral or photoelectrons as well as by soft solar X-rays .

5    The NO content is reduced by photodissociation

$$NO + h\nu \rightarrow N + O \quad (\lambda < 191\,\text{nm}), \tag{R3}$$

by photoionization

$$NO + h\nu \rightarrow NO^+ + e^- \quad (\lambda < 134\,\text{nm}), \tag{R4}$$

and by reacting with atomic nitrogen:

10   $$NO + N \rightarrow N_2 + O. \tag{R5}$$

 $N_2O$ has been retrieved in the mesosphere and thermosphere from MIPAS (see, e.g. Funke et al. (2008b, a)) and from Scisat-1/ACE-FTS (Sheese et al., 2016). Model–measurement studies by Semeniuk et al. (2008) attributed the source of this $N_2O$ to being most likely the reaction between $NO_2$ and N atoms produced by particle precipitation:

15   $$N + NO_2 \rightarrow N_2O + O. \tag{R6}$$

We note that photo-excitation and photolysis at 185 nm (vacuum UV) of NO or $NO_2$ mixtures in nitrogen, $N_2$, or helium mixtures at 1 atm leads to $N_2O$ formation (Maric and Burrows, 1992).  Both mechanisms explaining the production of $N_2O$ involve excited states of NO. Hence these pathways contribute to the loss of NO and potentially an additional daytime source of $N_2O$ in the upper atmosphere. $N_2O$ acts as an intermediate reservoir at high altitudes ($\gtrsim 90$ km, see Sheese et al.

20   (2016)), reacting with $O(^1D)$ in two well known channels to $N_2$ and $O_2$ as well as to 2NO. However, the largest $N_2O$ abundances are located below 60 km and originate primarily from the transport of tropospheric $N_2O$ into the stratosphere through the Brewer Dobson Circulation (Funke et al., 2008a, b; Sheese et al., 2016) but can reach up to 70 km in geomagnetic storm conditions (Funke et al., 2008a; Sheese et al., 2016). Both source and sink reactions indicate that NO behaves differently in sunlit conditions than in dark conditions. NO is produced  by particle precipitation at auroral latitudes,

25   but  in dark conditions (without photolysis) it is depleted only by reacting with atomic nitrogen (reaction (R5)). This asymmetry between production and depletion in dark conditions results in different lifetimes of NO.

Early work to parametrize NO in the lower thermosphere (100 km–150 km) used SNOE measurements from  03/1998–09/2000 et al., 2004). With these  two and a half years of data and using empirical orthogonal functions, the so-called NOEM (Nitric Oxide Empirical Model) estimates NO in the lower thermosphere as a function of the solar $f_{10.7\,\text{cm}}$ radio flux, the solar

30   declination angle, and the planetary Kp index. NOEM is still used as prior input for NO retrieval, for example from MIPAS (Bermejo-Pantaleón et al., 2011; Funke et al., 2012) and SCIAMACHY (Bender et al., 2017b) spectra. However,

two and a half years is relatively short compared to the 11-year solar cycle, and the years  1998 to 2000 encompass a period of elevated solar activity. To address this, a longer time series from AIM/SOFIE was used to determine the important drivers of NO in the lower thermosphere (90 km–140 km) by Hendrickx et al. (2017). Other recent work uses ten years of NO data from Odin/SMR from 85 km to 115 km (Kiviranta et al., 2018). Funke et al. (2016) derived a semi-empirical model of

5    $NO_y$ in the stratosphere and mesosphere from MIPAS data. Here we use Envisat/SCIAMACHY NO data from the nominal limb mode (Bender et al., 2017b, a). Apart from providing a similarly long time series of NO data, the nominal Envisat/SCIA-MACHY NO data cover the mesosphere from 60 km to 90 km (Bender et al., 2017b), bridging the gap between the stratosphere and lower thermosphere models.

     The manuscript is organized as follows: we present the data used in this work in Sect. 2. The two model variants, linear and

10   non-linear, are described in Sect. 3. Details about the parameter and uncertainty estimation are explained in Sect. 4, and we present the results in Sect. 5. Finally we conclude our findings in Sect. 6.

**2   Data**

**2.1   SCIAMACHY NO**

We use the SCIAMACHY (SCanning Imaging Absorption spectroMeter for Atmospheric CHartoghraphY) nitric oxide data

15   set version 6.2.1 (Bender et al., 2017a) retrieved from the nominal limb scan mode ($\approx$ 0–93 km). For a detailed instrument description, see Burrows et al. (1995); Bovensmann et al. (1999), and for details of the retrieval algorithm, see Bender et al. (2013, 2017b).

     The data were retrieved for the whole Envisat period (08/2002–04/2012). This satellite was orbiting in a sun-synchronous orbit at around 800 km altitude, with Equator crossing times of 10:00/22:00 local time. The NO number densities from the

20   SCIAMACHY nominal mode were retrieved from the NO gamma band emissions. Since those emissions are fluorescent emissions excited by solar UV, SCIAMACHY NO data are only available for the 10:00 dayside (downleg) part of the orbit. Furthermore, the retrieval was carried out for altitudes from 60 km to 160 km, but above approximately 90 km, the data reflect the scaled a priori densities from NOEM (Bender et al., 2017b). We therefore restrict the modelling to the mesosphere below 90 km.

25    We averaged the individual orbital data longitudinally on a daily basis according to their geomagnetic latitude within $10°$ bins. The geomagnetic latitude was determined according to the eccentric dipole approximation of the 12th generation of the International Geomagnetic Reference Field (IGRF12) (Thébault et al., 2015). In the vertical direction the original retrieval grid altitudes (2 km bins) were used. Note that mesospheric NO concentrations are related to geomagnetically as well as geographically based processes, but disentangling them is beyond the scope of the paper. Follow-up studies can build on the

30   method presented here and study, for example, longitudinally resolved timeseries.

     The measurement sensitivity is taken into account via the averaging kernel diagonal elements, and days where its binned average was below 0.002 were excluded from the timeseries. Considering this criterion, each bin (geomagnetic latitude and altitude) contains about 3400 data points.

**2.2 Proxies**

We use two proxies to model the NO number densities, one accounting for the  solar irradiance variations and one accounting for the  geomagnetic activity. Various proxies have been used or proposed to account for the  solar irradiance induced variations in mesospheric–thermospheric NO, which are in particular related to the eleven-year solar cycle. The NOEM (Nitric Oxide Empirical Model, Marsh et al. (2004)) uses the natural logarithm of the solar 10.7 cm radio flux $f_{10.7}$. More recent work on AIM/SOFIE NO (Hendrickx et al., 2017) uses the solar Lyman-$\alpha$ index because some of the main production and loss processes are driven by UV photons. Besides accounting for the long-term variation of NO with solar activity, the Lyman-$\alpha$ index also includes short-term UV variations and the associated NO production, for example caused by solar flares. Barth et al. (1988) have shown that the Lyman-$\alpha$ index directly relates to the observed NO at low latitudes (30°S–30°N). Thus we use it in this work as a proxy for NO.

In the same manner as for the  irradiance variations, the "right" geomagnetic index to model  particle-induced variations of NO  is a matter of  opinion. Kp is the oldest and most commonly used geomagnetic index, it was, for example, used in earlier work by Marsh et al. (2004) for modelling NO in the mesosphere and lower thermosphere. Kp is derived from magnetometer stations distributed at different latitudes and mostly in the northern hemisphere.  However, Hendrickx et al. (2015) found that the auroral electrojet index (AE) (Davis and Sugiura, 1966)  correlated better with SOFIE-derived NO concentrations (Hendrickx et al., 2015, 2017) (see also Sinnhuber et al., 2016). The AE index is derived from stations distributed almost evenly within the auroral latitude band. This distribution enables the AE index to be more closely related to the energy input into the atmosphere at these latitudes. Therefore, we use the auroral electrojet index (AE) as a proxy for geomagnetically induced NO. To account for the 10:00 satellite sampling, we average the hourly AE index from noon the day before to noon on the measurement day.

It should be noted that tests using Kp (or its linear equivalent Ap) instead of AE and using $f_{10.7}$ instead of Lyman-$\alpha$ suggested that the particular choice of index did not lead to significantly different results. Our choice of AE rather than Kp and Lyman-$\alpha$ over $f_{10.7}$ is physically based and motivated as described above.

**3 Regression model**

We denote the number density by $x_{\mathrm{NO}}$ as a function of the (geomagnetic, see Sect. 2.1) latitude $\phi$, the altitude $z$, and the time (measurement day) $t$: $x_{\mathrm{NO}}(\phi, z, t)$. In the following we often drop the subscript NO and combine the time direction into a vector $\boldsymbol{x}$ with the $i$th entry denoting the density at time $t_i$, such that $x_i(\phi, z) = x(\phi, z, t_i)$.

**3.1 Linear model**

In the (multi-)linear case, we relate the nitric oxide number densities $x_{\mathrm{NO}}(\phi, z, t)$ to the two proxies, the solar Lyman-$\alpha$ index (Ly$\alpha(t)$) and the geomagnetic AE index (AE$(t)$). Harmonic terms with $\omega = 1\,\mathrm{a}^{-1} = (365.25\,\mathrm{d})^{-1}$ account for annual and semiannual variations. The linear model, including a constant offset for the background density, describes the NO density according to Eq. (1):

$$x_{\mathrm{NO}}(\phi, z, t) = a(\phi, z) + b(\phi, z) \cdot \mathrm{Ly}\alpha(t) + c(\phi, z) \cdot \mathrm{AE}(t)$$
$$+ \sum_{n=1}^{2} [d_n(\phi, z) \cos(n\omega t) + e_n(\phi, z) \sin(n\omega t)] \ . \tag{1}$$

The linear model can be written in matrix form for the $n$ measurement times $t_1, \ldots, t_n$ as Eq. (2), with the parameter vector $\boldsymbol{\beta}$ given by $\boldsymbol{\beta}_{\mathrm{lin}} = (a, b, c, d_1, e_1, d_2, e_2)^\top \in \mathbb{R}^7$ and the model matrix $\mathbf{X} \in \mathbb{R}^{n \times 7}$. We determine the coefficients via least squares,

$$\boldsymbol{x}_{\mathrm{NO}}(\phi, z) = \begin{pmatrix} 1 & \mathrm{Ly}\alpha(t_1) & \mathrm{AE}(t_1) & \cos(\omega t_1) & \sin(\omega t_1) & \cos(2\omega t_1) & \sin(2\omega t_1) \\ \vdots & & & & & & \vdots \\ 1 & \mathrm{Ly}\alpha(t_n) & \mathrm{AE}(t_n) & \cos(\omega t_n) & \sin(\omega t_n) & \cos(2\omega t_n) & \sin(2\omega t_n) \end{pmatrix} \cdot \begin{pmatrix} a \\ b \\ c \\ d_1 \\ e_1 \\ d_2 \\ e_2 \end{pmatrix} \tag{2}$$

$$= \mathbf{X} \cdot \boldsymbol{\beta}$$

minimizing the squared differences of the modelled number densities to the measured ones.

**3.2 Non-linear model**

In contrast to the linear model above, we modify the AE index by a finite lifetime $\tau$ which varies according to season, we denote this modified version by $\widetilde{\mathrm{AE}}$. We then omit the harmonic parts in the model, and the non-linear model is given by Eq. (3):

10 $\quad x_{\mathrm{NO}}(\phi, z, t) = a(\phi, z) + b(\phi, z) \cdot \mathrm{Ly}\alpha(t) + c(\phi, z) \cdot \widetilde{\mathrm{AE}}(t) \ . \tag{3}$

Although this approach shifts all seasonal variations to the AE index and thus attributes them to particle-induced effects, we found that the residual traces of particle-unrelated seasonal effects were minor compared to the overall improvement of the fit. Additional harmonic terms increase only the number of free parameters without substantially improving the fit further.

The lifetime-corrected $\widetilde{\mathrm{AE}}$ is given by the sum of the previous $60\,\mathrm{days}$' AE values, each multiplied by an exponential decay

15 factor:

$$\widetilde{\mathrm{AE}}(t) = \sum_{t_i=0}^{60\,\mathrm{d}} \mathrm{AE}(t - t_i) \cdot \exp\left\{-\frac{t_i}{\tau}\right\} \ . \tag{4}$$

The total lifetime $\tau$ is given by a constant part $\tau_0$ plus the non-negative fraction of a seasonally varying part $\tau_t$:

$$\tau = \tau_0 + \begin{cases} \tau_t , & \tau_t \geq 0 \\ 0 , & \tau_t < 0 \end{cases} , \tag{5}$$

$$\tau_t = d\cos(\omega t) + e\sin(\omega t) . \tag{6}$$

$\tau_t$ accounts for the different lifetime during winter and summer. The parameter vector
for this model is given by $\boldsymbol{\beta}_{\mathrm{nonlin}} = (a, b, c, \tau_0, d, e)^\top \in \mathbb{R}^6$, and we describe how we determine these coefficients and their uncertainties in the next section.

**4   Parameter and uncertainty estimation**

The parameters are usually estimated by maximizing the likelihood, or, in the case of additional prior constraints, by maximizing the posterior probability. In the linear case and in the case of independently identically distributed Gaussian measurement
uncertainties, the maximum likelihood solutions are given by the usual linear least squares solutions. Estimating the parameters in the non-linear case is more involved. Various methods exist, for example conjugate gradient, random (Monte-Carlo) sampling or exhaustive search methods. The assessment and selection of the method to estimate the parameters in the non-linear case is given below.

**4.1   Maximum posterior probability**

Because of the complicated structure of the model function Eq. (3), in particular the lifetime parts in Eqs. (5) and (6), the usual gradient methods converge slowly, if at all. Therefore, we fit the parameters and assess their uncertainty ranges using Markov-Chain Monte-Carlo (MCMC) sampling (Foreman-Mackey et al., 2013). This method samples probability distributions and we apply it to sample the parameter space putting emphasis on parameter values with a high posterior probability. The posterior distribution is given in the Bayesian sense as the product of the likelihood and the prior distribution:

$$p(\boldsymbol{x}_{\mathrm{mod}}|\boldsymbol{y}) \propto p(\boldsymbol{x}_{\mathrm{mod}}|\boldsymbol{y}, \boldsymbol{\beta}) p(\boldsymbol{\beta}) . \tag{7}$$

We denote the vector of the measured densities by $\boldsymbol{y}$ and the modelled densities by $\boldsymbol{x}_{\mathrm{mod}}$ similar to Eqs. (1) and (3). To find the best parameters $\boldsymbol{\beta}$ for the model, we maximize $\log p(\boldsymbol{x}_{\mathrm{mod}}|\boldsymbol{y})$.

The likelihood $p(\boldsymbol{x}_{\mathrm{mod}}|\boldsymbol{y}, \boldsymbol{\beta})$ is in our case given by a Gaussian distribution of the residuals, the difference of the model to the data, Eq. (8). Note that the normalization constant $C$ in Eq. (8) does not influence the value of the maximal likelihood. The

$$p(\boldsymbol{x}_{\mathrm{mod}}|\boldsymbol{y}, \boldsymbol{\beta}) = \mathcal{N}(\boldsymbol{y}, \mathbf{S}_y) = C\exp\left\{ -\frac{1}{2}\left(\boldsymbol{y} - \boldsymbol{x}_{\mathrm{mod}}(\boldsymbol{\beta})\right)^\top \mathbf{S}_y^{-1}\left(\boldsymbol{y} - \boldsymbol{x}_{\mathrm{mod}}(\boldsymbol{\beta})\right) \right\} \tag{8}$$

**Table 1.** Parameter search space for the non-linear model and uncertainty estimation.

[revised manuscript text omitted]

**5.1 Time series fits**

The fitted densities of the linear model Eq. (1) compared to the data are shown in the upper panels of Fig 1 for the three example latitude bins (65°S, 5°N, 65°N) at 70 km. The linear model works well at high southern and low latitudes. At high northern latitudes and to a lesser extent at high southern latitudes, the linear model captures the summer NO variations well. However, the model underestimates the high values in the polar winter at active times (2004–2007) and overestimates the low winter values at quiet times (2009–2011).

For the sample timeseries (65°S, 5°N, 65°N at 70 km), the fits using the non-linear model Eq. (3) are shown in the upper panels of Fig 2. The non-linear model better captures both the summer NO variations as well as the high values in the winter, especially at high northern latitudes. However, at times of high solar activity (2003–2006) and in particular at times of a strongly disturbed mesosphere (2004, 2006, 2012), the residuals are still significant. At high southern and low latitudes, the improvement over the linear model is less evident. At low latitudes, the NO content is apparently mostly related to the eleven-year solar cycle and the particle influence is suppressed. Since this cycle is covered by the Lyman-$\alpha$ index, both models perform similarly, but the non-linear version has one less parameter. In both regions the residuals show traces of seasonal variations that are not related to particle effects. The linear model appears to capture these variations better than the non-linear model. However,

[Figure]

**Figure 1.** Time series data and linear model values and residuals at 70 km for 65°S (left), 5°N (middle), and 65°N (right). The top row shows the data (black dots with $2\sigma$ error bars) and the model values (blue line). The bottom row shows the residuals as black dots with $2\sigma$ error bars.

by objective measures including the number of model parameters[1], the non-linear version fits the data better in all bins (not shown here). At high southern latitudes, the SCIAMACHY data are less densely sampled compared to high northern latitudes (see Bender et al. (2017b)). In addition to the sampling differences, geomagnetic latitudes encompass a wider geographic range in the Southern Hemisphere (SH) than in the Northern Hemisphere (NH), and the AE index is derived from stations in the Northern Hemisphere. Both effects can lower the NO concentrations that SCIAMACHY observes in the Southern Hemisphere particularly at the winter maxima. The lifetime variation that improves the fit in the NH is thus less effective in the SH.

**5.2 Parameter morphologies**

Using the non-linear model, we show the latitude–altitude distributions of the medians of the sampled Lyman-$\alpha$ and geomagnetic index coefficients in Fig. 3. The white regions indicate values outside of the 95% confidence region or whose sampled distribution has a skewness larger than 0.33. The MCMC method samples the parameter probability distributions. Since we require the geomagnetic index and constant lifetime parameters to be larger than zero  (see Table 1), these sampled distributions are sometimes skewed towards zero  even though the 95% credible region is still larger than zero.  Excluding heavily skewed distributions avoids those cases because the "true" parameter is apparently zero  .
* * *
[1] Past and recent research in model selection provides a number of choices on how to compare models objectively. The results are so-called information criteria which aim to provide a consistent way of how to compare models, most notably the "Akaike Information Criterion" (AIC, Akaike (1974)), the "Bayesian Information Criterion" or "Schwarz Criterion" (BIC or SIC, Schwarz (1978)), the "Deviance Information Criterion" (DIC, Spiegelhalter et al. (2002); Ando (2011)), or the "Widely Applicable Information Criterion" (WAIC, Watanabe (2010); Vehtari et al. (2016)). Alternatively, the "Standardized Mean Squared Error" (SMSE) or the "Mean Standardized Log-Loss" (MSLL) (Rasmussen and Williams, 2006, Ch. 2) give an impression of the quality of regression models with respect to each other.

[Figure]

**Figure 2.** Same as Fig. 1 for the non-linear model.

[Figure]

**Figure 3.** Latitude–altitude distributions of the fitted solar index parameter (Lyman-$\alpha$, left) and the geomagnetic index parameter (AE, right) from the non-linear model.

The Lyman-$\alpha$ parameter distribution shows that its largest influence is at middle and low latitudes between 65 km and 80 km. Another increase of the Lyman-$\alpha$ coefficient is indicated at higher altitudes above 90 km. The penetration of Lyman-$\alpha$ radiation decreases with decreasing altitude as a result of scattering and absorption by air molecules. On the other hand the concentration of air decreases with altitude. At this stage we have not an unambiguous explanation of this behaviour, but it may be related to reaction pathways as laid out by Pendleton et al. (1983) which would relate the NO concentrations to the $CO_2$ and $H_2O$ (or OH, respectively) profiles. The Lyman-$\alpha$ coefficients are all negative below 65 km. We also observe negative values at high northern latitude at all altitudes and at high southern latitudes above 85 km. These negative coefficients indicate that  NO photodissociation or conversion to other species outweighs its production via UV radiation  in those places. The north–south asymmetry may be related to sampling and the difference in illumination with respect to geomagnetic latitudes, see Sect. 5.1.

[Figure]

**Figure 4.** Latitude–altitude distributions of the fitted base lifetime $\tau_0$ (left) and the amplitude of the annual variation $|\tau_t|$ (right) from the non-linear model.

The geomagnetic influence is largest at high latitudes between $50°$ and $75°$ above about $65\,\mathrm{km}$. The AE coefficients peak at around $72\,\mathrm{km}$ and indicate a further increase above $90\,\mathrm{km}$. This pattern of the geomagnetic influence matches the one found in Sinnhuber et al. (2016). Unfortunately both increased influences above $90\,\mathrm{km}$ in Lyman-$\alpha$ and AE cannot be studied at higher latitudes due to a large a priori contribution to the data.

The latitude-altitude distribution of the lifetime parameters are shown in Fig. 4. All shown values are within the 95% confidence region. As for the coefficients above, we also exclude regions where the skewness was larger than 0.33. The constant part of the lifetime, $\tau_0$, is below 2 days in most bins, except for exceptionally large values ($> 10$ days) at low latitudes ($0$–$20°$N) between $68\,\mathrm{km}$ and $74\,\mathrm{km}$. Although we constrained the lifetime with an exponential prior distribution, these large values apparently resulted in a better fit to the data. One explanation could be that because of the small geomagnetic influence (the AE coefficient is small in this region), the lifetime is more or less irrelevant. The amplitude of the annual variation ($|\tau_t| = \sqrt{\tau_{\cos}^2 + \tau_{\sin}^2} = \sqrt{d^2 + e^2}$, see Eq. (6)) is largest at high latitudes in the Northern Hemisphere and at middle latitudes in the Southern Hemisphere. This difference could be linked to the geomagnetic latitudes which include a wider range of geographic latitudes in the Southern Hemisphere compared to the Northern Hemisphere. Therefore, the annual variation is less apparent in the Southern Hemisphere. The amplitude also increases with decreasing altitude  below 75 km at middle and high latitudes and with increasing altitude above that. The increasing annual variation at low altitudes can be the result of transport processes that are not explicitly treated in our approach. Note that the term *lifetime* is not a pure (photo)chemical lifetime, rather it indicates how long the AE signal persists in the NO densities. In that sense it combines the (photo)chemical lifetime with transport effects as discussed in Sinnhuber et al. (2016).

**5.3 Parameter profiles**

For three selected latitude bins in the Northern Hemisphere ($5°$N, $35°$N, and $65°$N) we present profiles of the fitted parameters in Fig. 5. The solid line indicates the median and the error bars indicate the 95% confidence region. As indicated in Fig. 3, the

[Figure]

**Figure 5.** Coefficient profiles of the solar index parameter (Lyman-$\alpha$, left, (a)), the geomagnetic index parameter (AE, middle, (b)), and the amplitude of the annual variation of the NO lifetime (right, (c)) at 5°N (green), 35°N (orange), and 65°N (blue). The solid line indicates the median and the error bars indicate the 95% confidence region.

solar radiation influence is largest between 65 km and 80 km. Its influence is also up to a factor of two larger at low and middle latitudes compared to high latitudes, where the coefficient differs significantly from zero only below 65 km and above 82 km. Similarly, the geomagnetic impact decreases with decreasing latitude by one order of magnitude from high to middle latitudes and at least a further factor of five to lower latitudes. The largest impact is around 70–72 km and possibly above 90 km at

5   high latitudes, and is approximately constant between 66 km and 76 km at middle and low latitudes. Note that the scale on the middle panel in Fig. 5 is logarithmic. The lifetime variation shows that at high latitudes, geomagnetically affected NO persists longer at winter times (the phase is close to zero for all altitudes at 65°N, not shown here). It persists up to 10 days longer between 85 km and 70 km and increasingly longer below, reaching 28 days at 60 km.

For the same latitude bins in the Southern Hemisphere (5°S, 35°S, and 65°S) we present profiles of the fitted parameters

10   in Fig. 6. Similar to the coefficients in the Northern Hemisphere (see Fig. 5), the solar radiation influence is largest between 65 km and 80 km and also up to a factor of two larger at low and middle latitudes compared to high latitudes. However, the Lyman-$\alpha$ coefficients at 65°S are significant below 82 km. Also the geomagnetic AE coefficients show a similar pattern in the Southern Hemisphere compared to the Northern Hemisphere, decreasing by orders of magnitude from high to low latitudes. Note that the AE coefficients at high latitudes are slightly lower than in the Northern Hemisphere, whereas the coefficients

15   at middle and low latitudes are slightly larger. This slight asymmetry was also found in the study by Sinnhuber et al. (2016) and may be related to AE being derived solely from stations in the Northern Hemisphere (Mandea and Korte, 2011). With respect to latitude, the annual variation of the lifetime seems to be reversed compared to the Northern Hemisphere, with almost no variation at high latitudes and longer persisting NO at low latitudes. A faster descent in the southern polar vortex may be responsible for the short lifetime at high southern latitudes. Another reason may be the mixture of air from inside and

20   outside of the polar vertex when averaging along geomagnetic latitudes since the 65°S geomagnetic latitude band includes

[Figure]

**Figure 6.** Coefficient profiles of the solar index parameter (Lyman-$\alpha$, left, (a)), the geomagnetic index parameter (AE, middle, (b)), and the amplitude of the annual variation of the NO lifetime (right, (c)) at 5°S (green), 35°S (orange), and 65°S (blue). The solid line indicates the median and the error bars indicate the 95% confidence region.

geographic locations from about 45°S to 85°S. A third possibility may be the exclusion of the Southern Atlantic Anomaly from the retrieval (Bender et al., 2013, 2017b) where presumably the particle-induced impact on NO is largest.

**5.4 Discussion**

The distribution of the parameters confirms our understanding of the processes producing NO in the mesosphere to the largest part. The Lyman-$\alpha$ coefficients are related to radiative processes such as production by UV or soft X-rays, either directly or via intermediary of photoelectrons. The photons are not influenced by Earth's magnetic field and the influence of these processes is largest at low latitudes and decreases towards higher latitudes. We observe negative Lyman-$\alpha$ coefficients below 65 km at all latitudes and at high northern latitudes above 80 km. These negative Lyman-$\alpha$ coefficients indicate that at high solar activity photodissociation by $\lambda < 191\,\mathrm{nm}$ photons, photoionization by $\lambda < 134\,\mathrm{nm}$ photons, or collisional loss and conversion to other species outweigh the production from higher energy photons ($< 40\,\mathrm{nm}$). At high southern latitudes these negative Lyman-$\alpha$ coefficients are not as pronounced as at high northern latitudes. As mentioned in Sect. 5.2, this north–south asymmetry may be related to sampling and the difference in illumination with respect to geomagnetic latitudes, see also Sect. 5.1.

The AE coefficients are largest at auroral latitudes as expected for the particle nature of the associated NO production. The  AE coefficient can be considered as an effective production rate modulated by all short-time ($\ll 1\,\mathrm{day}$) processes. To roughly estimate this production rate, we divided the coefficient of the (daily) AE by 86400 s which follows the approach in Sinnhuber et al. (2016). We find a maximum production rate of about $1\,\mathrm{cm^{-3}\,nT^{-1}\,s^{-1}}$  around 70–72 km. This production rate also agrees with the one estimated by Sinnhuber et al. (2016) by a super-posed epoch analysis of summertime NO. Comparing the NO production to the ionization rates from Verronen et al. (2013) from 01–03 Jan 2005 (assuming approximately 1 NO molecule per ion pair), our model overestimates the ionization derived from AE on these days. The AE values

of 105 nT, 355 nT, and 435 nT translate to 105, 355, and 435 NO molecules $cm^{-3} s^{-1}$, about 4 times larger than would be estimated from the ionization rates in Verronen et al. (2013) but agreeing with Sinnhuber et al. (2016). The factor of 4 may be related to the slightly different locations, around 60°N (Verronen et al., 2013) compared to around 65°N here and in Sinnhuber et al. (2016) where the ionization rates may be higher.

The associated constant part $\tau_0$ of the lifetime ranges from around 1 to around 4 days, except for large $\tau_0$ at low latitudes around 70 km. As already discussed in Sect. 5.2, these large lifetimes may be a side-effect of the small geomagnetic coefficients and more or less arbitrary. The magnitude is similar to what was found in the study by Sinnhuber et al. (2016) using only the summer data.

The annual variation of the lifetime is largest at high northern latitudes with a nearly constant amplitude of 10 days between 70 and 85 km. An empirical lifetime of 10 days at winter was used by Sinnhuber et al. (2016) to extend the NO predicted by the summer analysis to the larger values at winter. Here we could confirm that 10 days is a good approximation of the NO lifetime at winter but it varies with altitude. The altitude distribution agrees with the increasing photochemical lifetime at  large solar zenith angles (Sinnhuber et al., 2016, Fig. 7b). The larger values in our study are similarly related to transport and mixing effects which alter the observed lifetime. The small variation of the lifetime at high southern latitudes could be a sampling issue because SCIAMACHY observes only small variations there at winter (see Figs. 1 and 2). Note that the results (in particular the large annual variation) in the northernmost latitude bin should be taken with caution because this bin is sparsely sampled by SCIAMACHY and the large winter NO concentrations are actually absent from the data.

**6 Conclusions**

We propose an empirical model to estimate the NO density in the mesosphere (60 km–90 km) derived from measurements from SCIAMACHY nominal mode limb scans. Our model calculates NO number densities for geomagnetic latitudes using the solar Lyman-$\alpha$ index and the geomagnetic AE index. Two approaches were tested, a linear approach containing annual and semi-annual harmonics, and a non-linear version using a finite and variable lifetime for the geomagnetically induced variations. From our proposed models, the linear variant describes only part of the NO variations. It can describe the summer variations but underestimates the large number densities at winter times. The non-linear version derived from the SCIAMACHY NO data describes both variations using  an annually varying finite lifetime  for the particle-induced NO. However, in cases of dynamic disturbances of the mesosphere, for example in early 2004 or in early 2006, the indirectly enhanced NO (see, for example, Randall et al., 2007) is not captured by the model. These remaining variations are treated as statistical noise.

Sinnhuber et al. (2016) use a super-posed epoch analysis limited to the polar summer to model the NO data. Here we extend that analysis to use all available SCIAMACHY nominal mode NO data for all seasons. However, during summer the present results show comparable NO production per AE unit and similar lifetimes to the Sinnhuber et al. (2016) study.

The parameter distributions indicate in which regions the different processes are significant. We find that these distributions match our current understanding of the processes producing and depleting NO in the mesosphere (Funke et al., 2014a, b, 2016; Sinnhuber et al., 2016; Hendrickx et al., 2017; Kiviranta et al., 2018). In particular, the influence of Lyman-$\alpha$ (or solar UV

radiation in general) is largest at low and middle latitudes which is explained by the direct production of NO via solar UV or soft X-ray radiation (Barth et al., 1988, 2003). The geomagnetic influence is largest at high latitudes and is best explained by the production from charged particles that enter the atmosphere in the polar regions along the magnetic field.

A potential improvement would be to use actual measurements of precipitating particles instead of the AE index. Using measured fluxes could help to confirm our current understanding of how those fluxes relate to ionization (Turunen et al., 2009; Verronen et al., 2013) and subsequent NO production (Sinnhuber et al., 2016). Furthermore, including dynamical transport as for example in Funke et al. (2016), could improve our knowledge of the combined direct and indirect NO production in the mesosphere.

*Author contributions.* S.B. developed the model, prepared and performed the data analysis and set up the manuscript, M.S. provided input on the model and the idea of a variable NO lifetime. J.P.B. and P.J.E. contributed to the discussion and use of language. All authors contributed to the interpretation and discussion of the method and the results as well as to writing the paper.

*Code and data availability.* The SCIAMACHY NO data set used in this study is available at `https://zenodo.org/record/1009078` (Bender et al., 2017a). The python code to prepare the data (daily zonal averaging) and to perform the analysis is available at `https://zenodo.org/record/1401370` (Bender, 2018a) or at `https://github.com/st-bender/sciapy`. The daily zonal mean NO data and the sampled parameter distributions are available at `https://zenodo.org/record/1342701` (Bender, 2018b). The solar Lyman-$\alpha$ index data were downloaded from `http://lasp.colorado.edu/lisird/data/composite_lyman_alpha/`, the AE index data were downloaded from `http://wdc.kugi.kyoto-u.ac.jp/aedir/`, and the daily mean values used in this study are available within the aforementioned data set.

*Acknowledgements.* S.B. and M.S. thank the Helmholtz-society for funding part of this project under the grant number VH-NG-624. S.B. and P.J.E. acknowledge support from the Birkeland Center for Space Sciences (BCSS), supported by the Research Council of Norway under the grant number 223252/F50. The SCIAMACHY project was a national contribution to the ESA Envisat, funded by German Aerospace (DLR), the Dutch Space Agency, SNO, and the Belgium ministry. The University of Bremen as Principal Investigator has led the scientific support and development of the SCIAMACHY instrument as well as the scientific exploitation of its data products. This work was performed on the Abel Cluster, owned by the University of Oslo and Uninett/Sigma2, and operated by the Department for Research Computing at USIT, the University of Oslo IT-department. http://www.hpc.uio.no/